# Finding Safe Zones of Markov Decision Processes Policies

**Lee Cohen**
Tel-Aviv University

**Yishay Mansour**
Tel-Aviv University
Google Research

**Michal Moshkovitz**
Tel-Aviv University

## Abstract

Safety is essential for gaining trust in Markov Decision Process's policies. We suggest a new method to improve safety, using SAFEZONES. Given a policy, we define its SAFEZONE as a subset of states, such that most of the policy's trajectories are confined to this subset. A trajectory not entirely inside the SAFEZONE is potentially unsafe and should be examined. The quality of the SAFEZONE is parameterized by the number of states and the escape probability, i.e., the probability that a random trajectory will leave the subset. SAFEZONES are especially interesting when they have a small number of states and low escape probability. We study the complexity of finding optimal SAFEZONES, and show that in general, the problem is computationally hard. For this reason, we concentrate on computing approximate SAFEZONES. Our main result is a bi-criteria approximation algorithm which gives a factor of almost 2 approximation for both the escape probability and SAFEZONE size, using a polynomial size sample complexity.

## 1 Introduction

Most research in reinforcement learning (RL) deals with the problem of learning an optimal policy for some Markov decision process (MDP). One notable exception for that is Safe RL, that focuses on finding the best policy that meets safety requirements. Typically, these problems are handled by adjusting the objective to include safety requirements and then optimizing over it, or incorporating additional safety constraints to the exploration stage. Anomaly Detection is the problem of identifying patterns in data that do not correspond to what is expected, i.e., anomalies. Anomaly Detection addresses a variety of applications: cyber-security, fraud detection, failure detection, etc. (see Chandola et al. (2009) for survey).

In this paper, we introduce the SAFEZONE problem, a general approach for safe RL and anomaly detection that concentrates on a given policy rather than finding a policy that follows some predefined safety specifications and emphasizes entire trajectories in order to detect anomalies.

Consider a policy for a finite horizon MDP. The policy induces a Markov Chain (MC) on the MDP. Given a subset of states, we define the *escape probability* to be the probability that a random trajectory has at least one state outside this subset (hence the trajectory *escapes* it). A SAFEZONE is a subset of states whose quality is measured by its' size and escape probability (ideally, both are small). If a SAFEZONE has low escape probability, we consider it *safe*.

Trivial SAFEZONE solutions are the entire set of states (which has minimal escape probability of $0$ on the account of maximal size), and the empty set (which has minimal size but has maximal escape probability of $1$). We are interested to find SAFEZONE with a good tradeoff: namely a relatively small set size with small escape probability. More precisely, given a bound over the escape probability, $\rho > 0$, the goal of the learner is, using trajectory sampling, to find the smallest SAFEZONE with escape probability at most $\rho$. We address unknown environment, by which we mean no prior knowledge on the transition function or the policy used. The learner can only access

2022 Trustworthy and Socially Responsible Machine Learning (TSRML 2022) co-located with NeurIPS 2022.

random trajectories generated by the induced MC. For many applications, if there exists such a small SAFEZONE it is useful to find it.

Consider for example automatic robotic arm that assembles products. If something unusual happened during the assembly of a product, it might result in a malfunctioning product. In that case, the operator should be notified (anomaly detection). On the other hand, we would not like to call the operator too often. If we find a SAFEZONE, we can make sure that we notify the operator only in the rare events the production process (trajectory) escapes it. Furthermore, if the SAFEZONE is small, the manufacturer can potentially test the SAFEZONE states and verify their compliance, ensuring that the majority of products are well constructed for a significantly lower testing budget.

Another useful application is transportation design. For example, given data regarding bicycle commutes (not necessarily done on bicycle lanes) in a populated areas, pave bike lanes in the SAFEZONE, namely in a way that would accommodate popular commutes, from starting point to destination. Making cycling safer and more accessible would also promote it as a viable transportation option, which in turn benefits the environment Watkins et al. (2020).

We remark that finding a SAFEZONE alone does not suffice for safety; Rather, a nearly optimal SAFEZONE is a behavioral description that can be used for safety applications, such as safer cycling. As another example, efficient testing (of states within the SAFEZONE ) that "captures" most of the products' assembly process would improve safety.

Other motivations include imitation learning with compact policy representation. Namely, design a smaller state policy that preforms well for most cases but might be undefined on some states. In this case, trajectories that reach undefined states have zero reward, and such trajectories are captured by the escape probability. One natural application for that is creating a 'lite' version for a given software such as Microsoft's Windows Lite.

Our work can also be viewed through the lens of explainable RL, where the goal is to explain a specific policy. SAFEZONE is a new post-hoc explanation of the summarization type (Alharin et al., 2020). Going back to the bicycle example, a municipality could provide a convincing explanation to its community for the chosen design.

Our results are the following:

1. Introducing the SAFEZONE problem (Section 2), and some of its applications.
2. We explore naive approaches, namely greedy algorithms that select SAFEZONES based on state distributions and trajectory sampling. In addition, we show cases in which their solutions are far from optimal, either in terms of high escape probability or significantly larger set size (see Section 3).
3. We design FINDING SAFEZONE, an efficient approximation algorithm with provable guarantees. The algorithm returns a SAFEZONE which is slightly more than twice in terms of both the size and the escape probability compared to the optimal (see Section 4).

For brevity, some algorithms and (full) proofs are relegated to the appendix.

**Trajectory escaping.** The SAFEZONE problem deals escaping trajectories. In particular, given a SAFEZONE, a trajectory escapes it, no matter if only one of its states is outside the SAFEZONE or all of them. A related, yet very different problem, is that of minimizing a subset size, such that the expected number of states outside the set is minimized. This related problem, while significantly easier (as it is solved by returning the most visited states), does not apply to the applications we described earlier. In Section 3, we show that the solution for the SAFEZONE does not necessarily overlaps with the most visited states. Furthermore, simply returning states which appeared in trajectory samples could result in a set size far from optimal.

## 1.1 Related Work

MDPs have been studied extensively in the context of decision making in particular by the Reinforcement Learning (RL) community (see Puterman (1994) for a broad background on MDPs, and Sutton & Barto (2018) for background on reinforcement learning).

**Safe RL.** A related line of research is safe RL, where the learner's goal is to find the best policy that satisfies safety guarantees. The two main methodologies to handle such problems are: (1) altering the objective to include the safety requirement and optimizing over it, and (2) adding safety constraints

to the exploration part. See Pfrommer et al. (2021); Emam et al. (2021); Xu et al. (2021); Hendrycks et al. (2021); HasanzadeZonuzy et al. (2021) for recent works and García & Fernández (2015); Amodei et al. (2016) for surveys. In our work, the goal is not to find the optimal policy, but instead given a policy, finding its SAFEZONE. Moreover, the SAFEZONE problem is not characterized by specific requirements, and beyond the MDP, the solution could very much depend on the given policy.

**Imitation Learning.** In imitation learning, the learner observes a policy behaviour and wants to imitate it (see Hussein et al. (2017) for survey). Similar to imitation learning, we are given access to samples of a given policy. In contrast, rather than imitating the policy we find the policy's SAFEZONE, which is an important property of the policy.

**Approximate MDP equivalence.** Another related research line is that of finding an (almost) equivalent minimal model for a given MDP, where the goal is that the optimal policy on the (almost) equivalent model induces an (approximately) optimal policy in the original MDP, e.g., Givan et al. (2003); Even-Dar & Mansour (2003). This line of works and ours differ in that we do not try to modify the MDP (e.g., cluster similar states), but rather to find a SAFEZONE, a property which is defined for the existing MDP and a specific policy.

**Explainability.** In explainability, the goal is to provide a post-hoc explanation to a specific given model Molnar (2019), e.g., using decision trees Blanc et al. (2021); Moshkovitz et al. (2021), influential examples Koh & Liang (2017), or a local approximation explanations Li et al. (2020). We focus on explainability for reinforcement learning, and specifically we suggest a new summarization explanation through our SAFEZONE, Amir & Amir (2018).

**MC with traps.** A decision problem that might seem related to ours is that of MC with traps (den Hollander et al. (1995)): Given an input of a MC (with possibly infinite state space), a starting state, and states trapping (absorbing) probabilities, the goal is to decide whether or not a (possibly infinite) random walk would reach an absorbing state with probability 1, or not.

## 2 The Safe Zone Problem

We model the problem using a Markov model with finite horizon $H > 1$. Formally, there is a Markov chain (MC) $\langle \mathcal{S}, P, s_0 \rangle$ where $\mathcal{S}$ is the set of states, $s_0 \in \mathcal{S}$ is the initial state and $P : \mathcal{S} \times \mathcal{S} \to [0, 1]$ is the transition function that maps a pair of states into probability by $P(s, s') = \Pr[s_{t+1} = s' | s_t = s]$. We assume the transition function $P$ is induced by a policy $\pi : \mathcal{S} \to \text{Simplex}^{\mathcal{A}}$ on an MDP $\langle \mathcal{S}, s_0, P', \mathcal{A} \rangle$ with transition function $P' : \mathcal{S} \times \mathcal{A} \times \mathcal{S} \to [0, 1]$ such that $P(s, s') = \sum_{a \in \mathcal{A}} P'(s, a, s') \cdot \pi(a|s)$ for all $s, s' \in \mathcal{S}$ (though any MC can be generated this way, thus our theoretical guarantees apply for general MCs).

A *trajectory* $\tau = (s_0, \ldots, s_H)$ starts in the initial state $s_0$ and followed by a sequence of $H$ states generated by $P$, i.e., $\Pr[s_{i+1} = s' | s_i = s] = P(s, s')$ for all $i \in [H]$, where $[H] := \{1, \ldots, H\}$. We abuse the notation and regard a trajectory $\tau$ both as a sequence and a set.

Given a subset of states $F \subseteq \mathcal{S}$, a trajectory $\tau$ *escapes* $F$ if it contains at least one state $s \in \tau$ such that $s \notin F$, i.e., $\tau \nsubseteq F$. We refer to the probability that a random trajectory escapes $F$ as *escape probability* and denote it by $\Delta(F) = \Pr_\tau[\tau \nsubseteq F]$. We call $F$ a $\rho-safe$ (w.r.t. the model $\langle \mathcal{S}, s_0, P \rangle$) if its escape probability, $\Delta(F)$, is at most $\rho$. Formally,

**Definition 2.1.** *A set $F \subseteq \mathcal{S}$ is $\rho-safe$ if*

$$\Delta(F) := \Pr_\tau[\tau \nsubseteq F] \leq \rho,$$

*where $\tau$ is a random trajectory.*

A set $F \subseteq \mathcal{S}$ is called $(\rho, k)-$SAFEZONE if $F$ is $\rho-$safe and $|F| \leq k$. Given a safety parameter $\rho \in (0, 1)$, we denote the smallest size $\rho-$safe set by $k^*(\rho)$:

$$k^*(\rho) = \min_{F \subseteq \mathcal{S} \text{ is } \rho-\text{safe}} |F|.$$

Whenever the discussed parameter $\rho$ is clear from the context we use $k^*$ instead of $k^*(\rho)$. We remark that there might be multiple different $(\rho, k)-$SAFEZONE sets.

The learner knows the set of states, $\mathcal{S}$, the initial state, $s_0$, and the horizon $H$ but has no knowledge regarding the transition function $P$ or the minimal size of the $\rho-$safe set, $k^*$. Instead, the learner receives information about the model from sampling trajectories from the distribution induced by $\pi$.

Table 1: Upper bounds for safety and size. * Only for layered MDPs.

| ALGORITHM | SAFE | SET SIZE | SAMPLE COMPLEXITY |
|---|---|---|---|
| GREEDY BY THRESHOLD | $2\rho$ | $k^* H/\rho$ | – |
| SIMULATION | $2\rho$ | $O(k^* H \ln k^*)$ | $poly(k^*, \frac{1}{\rho})$ |
| GREEDY AT EACH STEP* | $\rho H$ | $k^*$ | – |
| FINDING SAFEZONE | $2\rho + 2\epsilon$ | $(2 + \delta)k^*$ | $poly(k^*, H, \frac{1}{\epsilon}, \frac{1}{\delta})$ |

Given $\rho > 0$, the ultimate goal of the learner would have been to find a $(\rho, k^*(\rho))-$SAFEZONE. However, finding a $(\rho, k^*(\rho))-$SAFEZONE is NP-hard, even when the transition function $P$ is known. This is why we loosen the objective to find a bi-criteria approximation $(\rho', k')-$SAFEZONE . (Bi-criteria approximations are widely studied in approximation and online algorithms Vazirani (2001); Williamson & Shmoys (2011).) In our setting, given $\rho$ the objective is to find a set $F$ which is $(\rho', k')-$SAFEZONE with minimal size $k' \geq k^*$ and minimal escape probability $\rho' \geq \rho$. In addition, we are interested in minimizing the sample complexity.

Notice that the learner can efficiently verify, with high probability, whether a set $F$ is approximately $\rho-$safe or not.

**Summary of Contributions.** We summarize the results of all the algorithms that appear in the paper in Table 1. The bounds of GREEDY BY THRESHOLD and GREEDY AT EACH STEP requires the Markov Chain model as input, and a pre-processing step that takes $O(|S|^2 H)$ time. Additionally, the bounds for first three algorithms (the naive approaches) requires an additional knowledge of $k^*(\rho)$. Beyond the upper bounds, we provide instances that show that the upper bounds are tight up to a constant for each of the first three algorithms (the naive approaches). The following theorem is an informal statement of our main theorem.

**Theorem 2.2.** *For every $\rho, \epsilon, \delta > 0$, with probability $\geq 0.99$ there exists an algorithm that returns a set which is $(2\rho + 2\epsilon, (2 + \delta)k^*) -$ SAFEZONE.*

The running time of the algorithm is also bounded by $poly(k^*, H, \frac{1}{\delta}, \frac{1}{\epsilon})$.

## 3 Gentle Start

This section explains and analyzes various naive algorithms to the SAFEZONE problem. We show that even if the transition function is known in advance, these naive algorithms result in outputs that are far from optimal. To describe the algorithms, we define for each state $s$ the probability to appear in a random trajectory and denote it by $p(s) = \Pr_\tau[s \in \tau] \in [0, 1]$. Note that $\sum_{s \in S} p(s)$ is a number between 1 and $H$ (e.g., $p(s_0) = 1$), and can be estimated efficiently using dynamic programming if the environment and policy are known and sampling otherwise. To be precise, some of the algorithms assume the probabilities $\{p(s)\}_{s \in S}$ are received as input.

**Greedy by Threshold Algorithm.** The algorithm gets, in addition to $\rho$, the distribution $p$ and a parameter $\beta > 0$ as input. It returns a set $F$ that contains all states $s$ with probability at least $\beta$, i.e., $p(s) \geq \beta$. We formalize this idea as Algorithm 3 in Appendix A. For $\beta = \frac{\rho}{k^*}$, the output of the algorithm is $\left(2\rho, \frac{k^* H}{\rho}\right) -$ SAFEZONE. More generally, we prove the following lemma.

**Lemma 3.1.** *For any $\rho, \beta \in (0, 1)$, the GREEDY BY THRESHOLD ALGORITHM returns a set that is $(\rho + k^*\beta, \frac{H}{\beta}) -$ SAFEZONE. In particular, for $\beta = \frac{\rho}{k^*}$, this set is $\left(2\rho, \frac{k^* H}{\rho}\right) -$ SAFEZONE.*

While it is clear why there are instances for which the safety is tight, Lemma A.1 in Appendix A shows that the set size is tight as well.

**Simulation Algorithm.** The algorithm samples $O(\frac{\ln k^*}{\beta})$ random trajectories and returns a set $F$ with all the states in theses trajectories. It is formalized in Appendix A as Algorithm 4.

**Lemma 3.2.** *Fix $\rho, \beta \in (0,1)$. With probability at least $0.99$,* SIMULATION *Algorithm returns a set that is* $\left(\rho + k^* \beta, O(k^* + \frac{\rho H \ln k^*}{\beta})\right) -$ SAFEZONE. *In particular, for $\beta = \frac{\rho}{k^*}$, this set is* $(2\rho, O(k^* H \ln k^*)) -$ SAFEZONE.

While this algorithm achieves a low escape probability, only $2\rho$, in Lemma A.2 in the appendix we prove that the size of $F$ is tight up to a constant, i.e., an MDP instance where $|F| = \Omega(k^* H \ln k^*)$.

So far, the presented algorithms were approximately safe (i.e., low escape probability), but might return large subsets. Without further assumptions, the following algorithm provides a $(\rho H, H k^*)-$SAFEZONE. However, when considering MDPs with a special structure it provides an optimal sized SAFEZONE , at the price of large escape probability.

**Greedy at Each Step Algorithm.** For the analysis of the next algorithm we assume the MDP is *layered*, i.e., there are no states that appear in more than a single time step and denote $\mathcal{S} = \bigcup_{i=1}^{H} \mathcal{S}_i$. I.e., the transitions $P(s, s')$ are nonzero only for $s' \in \mathcal{S}_{i+1}$ and $s \in \mathcal{S}_i$. The GREEDY AT EACH STEP ALGORITHM takes at each time step $i$ the minimal number of states such that the sum of their probabilities is at least $1 - \rho$. It is formalized in Appendix A as Algorithm 5.

**Lemma 3.3.** *For any $\rho \in (0,1)$, if the MDP is layered,* GREEDY AT EACH STEP ALGORITHM *returns a set that is $(\rho H, k^*) -$ SAFEZONE.*

In Lemma A.3 we have a lower bound on the escape probability, which asymptotically matches.

**Weaknesses of the naive algorithms.** We showed algorithms that identify SAFEZONE with either escape probability much greater than $\rho$ or with size much greater than $k^*$. This holds even when providing extra information (such as the transition function and/or the optimal size of the $\rho-$safe set, i.e., $k^*$). Moreover, we showed tight lower bounds for these algorithms.

# 4 Algorithm for Detecting Safe Zones

In this section we suggest a new algorithm that builds upon and improves the added trajectory selection of the SIMULATION Algorithm. One reason for why SIMULATION returns a large set is that it treats every sampled trajectory identically, regardless of how many states are being added.

More precisely, fix any $(\rho, k^*)-$SAFEZONE set, $F^*$, and consider a trajectory $\tau$ that escapes it, i.e., $\tau \nsubseteq F^*$. If $\tau$ was sampled, its states are added to the constructed set $F$, which might increase the size of $F$ by up to $H$ states that are not in $F^*$, without significantly improving the safety.

In contrast, when selecting which trajectory to add to $F$, we would consider the number of states it adds to the current set. For the sake of readability, we refer to any state which is not in the current set $F$ as *new*, and denote by $new_F(\tau)$ the number of new states in $\tau$ w.r.t. $F$, i.e.,

$$new_F(\tau) := |\tau \setminus F|.$$

Note that for every $F \subseteq \mathcal{S}$, we have that $\Pr_\tau[new_F(\tau) \neq 0] = \Delta(F)$.

The new algorithm does not sample each trajectory uniformly at random, but sample from a new distribution, which will be denoted by $Q_F$. While favoring trajectories with higher probabilities, which we already get by the sampling process, another key idea would guide this new distribution: To prefer trajectories that *gradually* increase the size of $F$. To implement this idea, we will ensure that the probability of adding a trajectory $\tau$ to $F$ should be *inversely proportional* to $new_F(\tau)$.

Formally, the support of $Q_F$ is the trajectories with new states, i.e., $X = \{\tau | new_F(\tau) \neq 0\}$. For every $\tau \in X$, $Q_F(\tau) \propto \frac{\Pr[\tau]}{new_F(\tau)}$, where $\Pr[\tau]$ is the probability of trajectory $\tau$ under the Markov Chain with dynamics $P$. Note that the new distribution depends on the current set $F$, and changes as we modify it. Intuitively, adding trajectories to $F$ according to $Q_F$ instead of adding trajectories sampled directly from the dynamics (as we do in SIMULATION) would increase the expected ratio between the added safety and the number of new states we add to $F$, thus improving the set size guarantee of the output set. We elaborate on this in Appendix B.1.2.

Our main algorithm is FINDING SAFEZONE, Algorithm 1. The algorithm receives, in addition to the safety parameter $\rho$, parameters $\epsilon, \lambda \in (0,1)$, and maintains a set $F$ that is initiated to $\{s_0\}$. On a high level, to implement the idea of adding trajectories to $F$ according to $Q_F$, we use *rejection sampling*.

Namely, in each iteration of the while–loop we first sample a trajectory $\tau$ and if $new_F(\tau) \neq 0$, we *accept* it with probability $1/new_F(\tau)$. If the trajectory is accepted, it is added to $F$. More precisely, if $new_F(\tau) \neq 0$, we sample a Bernoulli random variable, $accept \sim Br(1/new_F(\tau))$. If $accept = 1$, we add $\tau$ to $F$. This process of adding trajectories to $F$ generates the desired distribution, $Q_F$.

Whenever a trajectory is added to $F$, we estimate the escape probability $\Delta(F)$ (w.r.t. the updated set, $F$). The algorithm stops adding states to $F$ and returns it as output when it becomes "safe enough". To be precise, let $\widehat{\Delta}(F)$ denote the result of the escape probability estimation (by sampling trajectories and returning the fraction of trajectories escaping $F$). If $\widehat{\Delta}(F) \leq 2\rho + \epsilon$, it means that $F$ is $(2\rho + 2\epsilon)-$safe with probability $\geq 1 - \lambda_j > 1 - \lambda$, in which case the algorithm terminates and returns $F$ as output. To implement the estimation $\widehat{\Delta}(F)$, the algorithm calls *EstSafety* Subroutine. The subroutine samples $N_j = \Theta(\frac{1}{\epsilon^2} \ln \frac{2}{\lambda_j})$ trajectories, and returns the fraction of trajectories that escaped $F$.

---

| **Algorithm 1** FINDING SAFEZONE | **Algorithm 2** *EstSafety* Subroutine |
|---|---|
| Input: $\rho \in (0,1)$ | Input: subset $F$ |
| Parameters: $\epsilon, \lambda \in (0,1)$ | Parameters: $\epsilon, \lambda_j \in (0,1)$ |
| $F \leftarrow \{s_0\}, j \leftarrow 1, \widehat{\Delta}(F) \leftarrow 1$ | $\widehat{\Delta}(F) \leftarrow 0$ |
| **while** $\widehat{\Delta}(F) > 2\rho + \epsilon$ **do** | $\mathcal{T} \leftarrow$ sample $N_j = \frac{1}{2\epsilon^2} \ln \frac{2}{\lambda_j}$ trajectories |
| $\quad \tau \leftarrow$ sample a random trajectory | **for** $\tau \in \mathcal{T}$ **do** |
| $\quad$ Compute $new_F(\tau)$ | $\quad$ **if** $\tau \not\subseteq F$ **then** |
| $\quad$ **if** $new_F(\tau) \neq 0$ **then** | $\quad\quad \widehat{\Delta}(F) \leftarrow \widehat{\Delta}(F) + \frac{1}{N_j}$ |
| $\quad\quad$ sample $accept \sim Br(1/new_F(\tau))$ | $\quad$ **end if** |
| $\quad\quad$ **if** $accept = 1$ **then** | **end for** |
| $\quad\quad\quad F \leftarrow F \cup \tau$ | return $\widehat{\Delta}(F)$ |
| $\quad\quad\quad \lambda_j \leftarrow \frac{3\lambda}{2(j\pi)^2}, j \leftarrow j + 1$ | |
| $\quad\quad\quad \widehat{\Delta}(F) \leftarrow EstSafety(\epsilon, \lambda_j, F)$ | |
| $\quad\quad$ **end if** | |
| $\quad$ **end if** | |
| **end while** | |
| return $F$ | |

---

## 5  Discussion and Open Problems

In this paper, we have introduced the SAFEZONE problem. We designed a nearly $(2\rho, 2k^*)$ approximation algorithm for the case where the model and policy are unknown to the algorithm. Beyond improving the approximation factors (or showing that it cannot be done unless $P = NP$), a natural direction for future work is the following. Given $\rho > 0$ and an MDP (known or unknown to the learner), find a policy with a small $\rho-$safe set, with nearly optimal value. In fact, an efficient solution for this could pave the way to improve compactness of the policy representation. An interesting observation that comes up from the empirical demonstration is that different policies result in different sizes of SAFEZONES, and that the optimal policy does not necessarily has the smallest SAFEZONE.

## Acknowledgement

LC is supported in part by the Ariane de Rothschild Women Doctoral Program. This project has received funding from the European Research Council (ERC) under the European Union's Horizon 2020 research and innovation program (grant agreement No. 882396), by the Israel Science Foundation (grant number 993/17), Tel Aviv University Center for AI and Data Science (TAD), and the Yandex Initiative for Machine Learning at Tel Aviv University.

## Acknowledgement

LC was supported in part by the Ariane de Rothschild Women Doctoral Program. This project has received funding from the European Research Council (ERC) under the European Union's Horizon 2020 research and innovation program (grant agreement No. 882396), by the Israel Science Foundation (grant number 993/17), Tel Aviv University Center for AI and Data Science (TAD), and the Yandex Initiative for Machine Learning at Tel Aviv University.

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

# A Proofs of Section 3

## A.1 Greedy by Threshold Algorithm

A naive approach to the SAFEZONE problem is to return all states $s \in \mathcal{S}$ with probability $p(s) \geq \beta$, for some parameter $\beta > 0$, see Algorithm 3.

---

**Algorithm 3** Greedy by Threshold

---

   Parameter: $\beta > 0, \{p(s)\}_{s \in \mathcal{S}}$
   return $\{s \in \mathcal{S} : p(s) \geq \beta\}$

---

**Lemma 3.1.** *For any $\rho, \beta \in (0, 1)$, the* GREEDY BY THRESHOLD ALGORITHM *returns a set that is* $(\rho + k^*\beta, \frac{H}{\beta}) -$ SAFEZONE. *In particular, for $\beta = \frac{\rho}{k^*}$, this set is* $\left(2\rho, \frac{k^*H}{\rho}\right) -$ SAFEZONE.

*Proof.* There are at most $\frac{H}{\beta}$ states with probability $p(s) \geq \beta$. Thus $|F| \leq \frac{H}{\beta}$.

Denote by $F^*$ the optimal $(\rho, k^*) -$ SAFEZONE set. By law of total probability,

$$\Pr_{\tau}[\tau \not\subseteq F] \leq \Pr_{\tau}[\tau \not\subseteq F^*] + \Pr_{\tau}[\tau \subseteq F^* \setminus F].$$

Looking at the R.H.S of the inequality, the left term is smaller than $\rho$ by the definition of SAFEZONE. The right term is equal to the probability to reach a state in $F^*$ that its probability is smaller than $\beta$, i.e., a state in $F^* \setminus F$.

Using union bound, this can be bounded by $k^*\beta$. $\qquad\square$

**Lemma A.1.** *For every $\rho \in (0, 1/2), H \in \mathbb{N}$, there exists an MDP and a minimal integer $k$ such that the MDP has a $(\rho, k) -$ SAFEZONE , but for $\beta = \rho/k$ GREEDY BY THRESHOLD Algorithm returns $F$ with escape probability $\leq 2\rho$ and of size $|F| = \Omega(H/\beta)$.*

*Proof.* Fix $\rho \in (0, 1)$. For ease of the presentation we will assume that $\frac{1-\rho}{\beta}$ is an integer (if not, it should be rounded to the nearest integer). Define $A$ to contain $\frac{1-\rho}{\beta} \cdot H$ states, $B$ to contain $k - 1$ states, and $\mathcal{S} = \{s_0\} \cup A \cup B$. Consider the following MDP with states $\mathcal{S}$ and starting state $s_0$. The transition function is defined as follows:

- For every $i \in A$, $\Pr[s_{1,i}^A|s_0] = \beta$ and for every $j \in [H-1]$, $\Pr[s_{j+1,i}^A|s_{j,i}^A] = 1$.

- For $s \in B$, $\Pr[s|s_0] = \frac{1-\rho}{k-1}$

- For $s \in B$, $\Pr[s|s] = 1$

The MDP is illustrated in Figure 1. Clearly, $\{s_0\} \cup B$ is a $(\rho, k) -$ SAFEZONE . In addition, GREEDY BY THRESHOLD ALGORITHM returns the set of all states, as for every state $s \in A$ we have that $p(s) = \beta, p(s_0) = 1 > \rho \geq \beta$, and for every $s \in B$ we have that $p(s) = \frac{1-\rho}{k-1} > \frac{\rho}{k} = \beta$. Thus the size of the returned set is $\mathcal{S}$, which is of size $\Omega(H/\beta)$, which completes the proof. $\qquad\square$

## A.2 Simulation Algorithm

**Lemma 3.2.** *Fix $\rho, \beta \in (0, 1)$. With probability at least $0.99$,* SIMULATION *Algorithm returns a set that is* $\left(\rho + k^*\beta, O(k^* + \frac{\rho H \ln k^*}{\beta})\right) -$ SAFEZONE. *In particular, for $\beta = \frac{\rho}{k^*}$, this set is* $(2\rho, O(k^* H \ln k^*)) -$ SAFEZONE.

*Proof.* Denote by $F^*$ the optimal $(\rho, k^*) -$ SAFEZONE set. By the law of total expectation, we can split $\mathbb{E}[|F|]$ into two parts, depending on whether trajectories are entirely in $F^*$ or not:

- Trajectories that are entirely in $F^*$ contribute at most $k^*$ states to $F$.

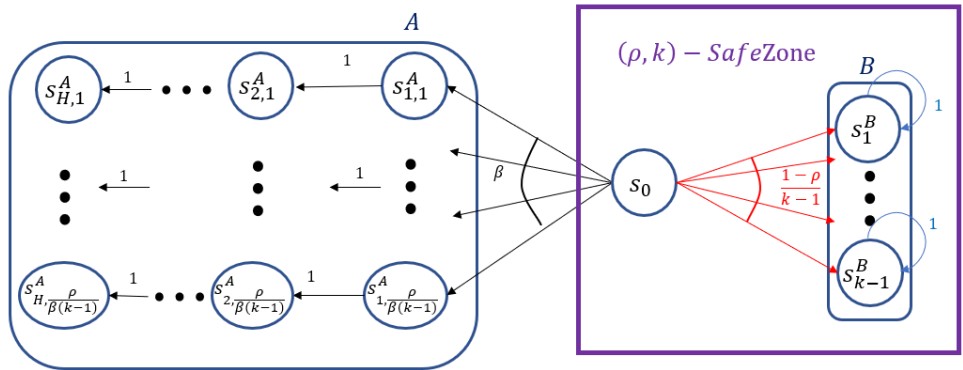

Figure 1: Lower bound for GREEDY BY THRESHOLD Algorithm.

---

**Algorithm 4** Simulation Algorithm

---

Input: $m = \frac{1}{\beta} \ln \frac{k^*}{0.005}$
$F \leftarrow \{s_0\}$
**for** $i = 1 \ldots m$ **do**
    $\tau \leftarrow$ choose a random trajectory
    $F \leftarrow F \cup \tau$
**end for**
return $F$

---

- A trajectory that is not contained in $F^*$ contributes at most $H$ states to $F$.

Thus,

$$\mathbb{E}[|F|] \le k^* + \rho \cdot \left( \frac{1}{\beta} \ln \frac{k^*}{0.005} \right) \cdot H = O\left( k^* + \frac{\rho H \ln k^*}{\beta} \right).$$

We use Markov's inequality to get the desired bound on $|F|$.

For the safety, we first denote the set of all states in $F^*$ with probability at least $\beta$ as $\Gamma = \{s \in F^* \mid p(s) \ge \beta\}$. We will show that with probability at least 0.9995, it holds that $\Gamma \subseteq F$, which will prove our claim, similarly to Lemma 3.1.

For a fixed state $s \in \Gamma$, the probability that $s \notin F$ is bounded by $(1 - p(s))^{\frac{1}{\beta} \ln \frac{k^*}{0.005}} \le e^{-\frac{\beta}{\beta} \cdot \ln \frac{k^*}{0.005}} = \frac{0.005}{k^*}$. Using union bound, the probability that there is a state $s \in \Gamma$ which is not in $F$ is bounded by $k^* \cdot \frac{0.005}{k^*} = 0.005$.

In other words, with probability at least 0.995, $\Gamma \subseteq F$, thus implementing the greedy approach in Algorithm 3 and proving that the probability that a random trajectory escapes $F$ is bounded by $\rho + k^* \beta$. □

**Lemma A.2.** *For every* $\rho, \gamma \in (0,1)$, $H, k \in \mathbb{N}$, *and* $\beta = \frac{\rho}{k}$, *there is an integer* $r \in \mathbb{N}$ *and MDP with* $(\rho, k)-$SAFEZONE, *but with probability* $\ge 1 - \gamma$, SIMULATION *algorithm returns $F$ of size* $\mathbb{E}[|F|] \ge kH \ln k$ *with escape probability* $\Delta(F) = O(\rho)$.

*Proof.* Fix $\rho, \gamma \in (0,1)$. Recall that $m = \frac{1}{\beta} \ln \frac{k^*}{0.005}$ and take $r = \lceil \frac{m^2}{\gamma} \rceil$. Define $A$ to contain $rH$ states, $B$ to contain $k-1$ states, and $\mathcal{S} = \{s_0\} \cup A \cup B$.

Consider the following MDP with states $\mathcal{S}$ and starting state $s_0$. The transition function is defined as follows:

- For every $i \in A$, $\Pr[s_{1,i}^A | s_0] = \frac{\rho}{r}$ and for every $j \in [H-1]$, $\Pr[s_{j+1,i}^A | s_{j,i}^A] = 1$.

- For $s \in B$, $\Pr[s | s_0] = \frac{1-\rho}{k-1}$

- For $s \in B$, $\Pr[s | s] = 1$

The MDP is illustrated in Figure 2.

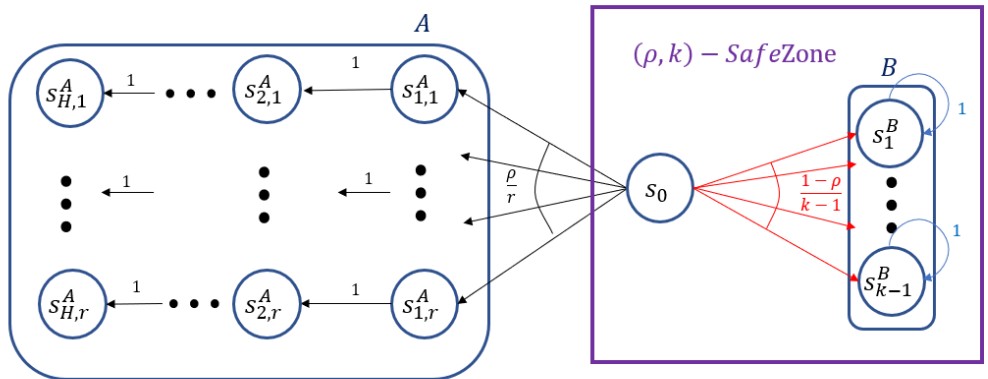

Figure 2: Lower bound for SIMULATION Algorithm.

The set $B \cup \{s_0\}$ is $\rho-$safe with $k$ states.

We will show that:

- After adding $\geq \frac{1}{\beta} \ln k = \frac{k}{\rho} \ln k$ random trajectories, with probability $\geq 1 - \gamma$ we have that $|F| \geq kH \ln k$.

- After adding $m$ random trajectories, we have that with high probability $F^* \subseteq F$, thus $\Delta(F) \leq \Omega(\rho)$.

To prove the first property, we claim that with probability$\geq 1 - \gamma$, every time we add a trajectory $\tau$ such that $\tau \cap A \neq \emptyset$, we add $H$ new states.

Notice that if we ignore $s_0$, trajectories in $A$ are entirely unconnected, and each trajectory is chosen randomly with probability $\Pr[s_{1,i}^A | s_0] = \frac{\rho}{r}$. This yields that if $s_{1,i}^A \notin F$, then $s_{j,i}^A \notin F$ for every $j \in [H]$. As a result, every time we add a new $s_{1,i}^A$ to $F$, we add $H-1$ more states to $F$. Let $N$ denotes the amount of trajectories sampled with states from $A$. The probability that their intersection contains only $s_0$ is

$$\frac{r \cdot (r-1) \cdot \ldots \cdot (r-N)}{r^N} \geq \left( \frac{r-N}{r} \right)^N = \left( 1 - \frac{N}{r} \right)^N \geq 1 - \frac{N^2}{r} = 1 - \gamma.$$

From the structure of the MDP, we have that $\mathbb{E}[N] = \rho m$. Therefore, with probability $\geq 1 - \gamma$,

$$\mathbb{E}[|F|] \geq \mathbb{E}[N] \cdot H = \rho \cdot m \cdot H \geq \rho \cdot \frac{1}{\beta} \ln k \cdot H = kH \ln k.$$

The second property follows from Lemma 3.2.

$\square$

### A.3 Greedy at Each Step

**Lemma 3.3.** *For any $\rho \in (0,1)$, if the MDP is layered,* GREEDY AT EACH STEP ALGORITHM *returns a set that is $(\rho H, k^*) -$ SAFEZONE.*

**Algorithm 5** Greedy at Each Step

Input: $\rho > 0$, $\{p(s)\}_{s \in \mathcal{S}}$
$F \leftarrow \{s_0\}$
**for** $i = 1 \dots H$ **do**
    Sort states in $\mathcal{S}_i$, $p(s_i^1) \geq \dots \geq p(s_i^{|\mathcal{S}_i|})$
    $j^* \leftarrow \arg\min_{j \in [|\mathcal{S}_i|]} \sum_{r=1}^{j} p(s_i^r) \geq 1 - \rho$
    $F \leftarrow F \cup \left\{ s_i^1, \dots s_i^{j^*} \right\}$
**end for**
return $F$

---

*Proof.* Take a random trajectory $\tau = (s_1, s_2, \dots)$. For every $s_i \in \tau$, the probability that $s_i \notin F$ is bounded by $\rho$, thus using union bound, the probability that $\tau$ has state $s_i$ such that $s_i \notin F$ is at most $\rho H$.

The construction of $F$ guarantees that $F$ is the minimal subset of states such that for every $i$, the probability that $s_i$ is in the subset is at least $1 - \rho$. Assume by contradiction that $|F| > k^*$. Then there is a time step $i$ such that $\Pr[s_i \in F^*] < 1 - \rho$, which is a contradiction, since $\Pr[\tau \in F^*] \leq \min_i \Pr[s_i \in F^*]$.

$\square$

**Lemma A.3.** *For any $\rho \in (0,1)$, there is an MDP and an integer $k$ such that there is a $(\rho, k)-$SAFEZONE , but* GREEDY AT EACH STEP *Algorithm returns $F$ with escape probability $\Delta(F) \geq \Omega(H\rho)$.*

*Proof.* Fix $\rho \in (0,1)$ and take $k = 3H + 1$.

Consider the MDP illustrated in Figure 3. The set $\{s_0\} \cup \{s_1^i\}_i \cup \{s_2^i\}_i \cup \{s_3^i\}_i$ form a $(\rho, 3H + 1)-$SAFEZONE .

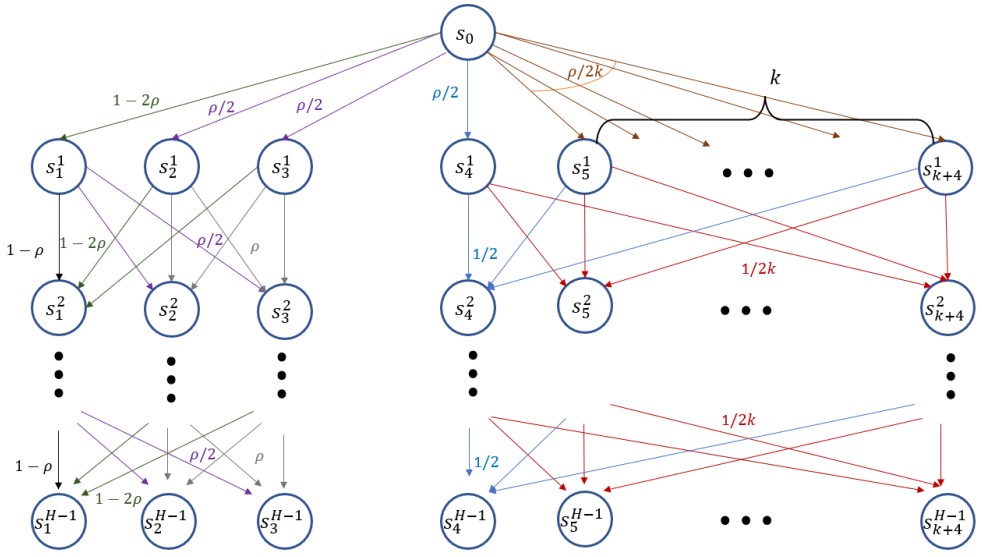

Figure 3: Lower bound for GREEDY AT EACH STEP Algorithm.

We will prove by induction that the for every time $i$,

- $p(s_1^i) = 1 - 2\rho$,
- $p(s_2^i) = p(s_3^i) = p(s_4^i) = \frac{\rho}{2}$, and

- For every $j \in \{5, \ldots, k+4\}$, $p(s_j^i) = \frac{\rho}{2k}$.

It is easy to see that the two properties hold for $i = 1$.

For $i > 1$,

$$p(s_1^i) = p(s_1^{i-1})(1-\rho) + p(s_2^{i-1})\frac{\rho}{2} + p(s_3^{i-1})\frac{\rho}{2} = (1-2\rho)(1-\rho) + 2(1-2\rho)\frac{\rho}{2} = 1 - 2\rho$$

$$p(s_2^i) = p_{i-1}(s_1^{i-1})\frac{\rho}{2} + p(s_2^{i-1})\rho + p(s_3^{i-1})\rho = (1-2\rho)\frac{\rho}{2} + \frac{\rho^2}{2} + \frac{\rho^2}{2} = \frac{\rho}{2}$$

Similarly, $p(s_3^i) = \frac{\rho}{2}$.

$$p(s_4^i) = \frac{1}{2}p(s_4^{i-1}) + \sum_{j=5}^{k+4}\frac{p(s_j^{i-1})}{2} = \frac{\rho}{4} + k\frac{\rho}{4k} = \frac{\rho}{2}$$

For every $j \in \{5, \ldots, k+4\}$,

$$p(s_j^i) = \frac{1}{2k}p(s_4^{i-1}) + \sum_{m=5}^{k+4}\frac{p(s_m^{i-1})}{2k} = \frac{\rho}{4k} + k\frac{\rho}{4k^2} = \frac{\rho}{2k}.$$

The algorithm might return $\{s_0\} \cup \{s_1^i\}_i \cup \{s_2^i\}_i \cup \{s_4^i\}_i$, i.e., instead of taking $\cup_i\{s_3^i\}_i$ it takes $\cup_i\{s_4^i\}_i$. Finally, the observation $\Delta(\{s_0\} \cup \{s_1^i\}_i \cup \{s_2^i\}_i \cup \{s_4^i\}_i) \geq \frac{\rho H}{4}$ completes the proof. □

# B    Proofs of Section 4

## B.1    High-level analysis

### B.1.1    Algorithm Analysis

We define the event

$$\mathcal{E} = \{\forall i \; |\widehat{\Delta}(F_{i-1}) - \Delta(F_{i-1})| \leq \epsilon\},$$

which states that all our *EstSafety* Subroutine estimations are accurate. We show that $\mathcal{E}$ holds with high probability using Hoeffding's inequality. In most of the analysis we condition on $\mathcal{E}$ to hold.

The following theorem is the central component in the proof of the main theorem that follows it.

**Theorem B.1.** *Given $\rho, \epsilon, \lambda \in (0,1)$, FINDING SAFEZONE Algorithm returns a subset $F \subseteq S$ such that:*

1. *The escape probability is bounded from above by $\Delta(F) \leq 2\rho + 2\epsilon$, with probability $1 - \lambda$.*
2. *The expected size of $F$ given $\mathcal{E}$ is bounded by $\mathbb{E}[|F| \mid \mathcal{E}] \leq 2k^*$.*
3. *The sample complexity of the algorithm is bounded by $O\left(\frac{k^*}{\lambda\epsilon^2}\ln\frac{k^*}{\lambda} + \frac{Hk^*}{\rho\lambda}\right)$, and the running time is bounded by $O\left(\frac{Hk^*}{\lambda\epsilon^2}\ln\frac{k^*}{\lambda} + \frac{H^2k^*}{\rho\lambda}\right)$, with probability $1 - \lambda$.*

To obtain the main theorem, we run FINDING SAFEZONE Algorithm several times and return the smallest output set, $F$, see the next section for more details.

**Theorem B.2.** *(main theorem) Given $\epsilon, \rho, \delta > 0$, if we run FINDING SAFEZONE for $\Theta(\frac{1}{\delta})$ times and return the smallest output set, $F \subseteq S$, then with probability $\geq 0.99$*

1. *The escape probability is bounded by $\Delta(F) \leq 2\rho + 2\epsilon$.*
2. *The size of $F$ is bounded from above by $|F| \leq (2+\delta)k^*$.*
3. *The total sample complexity and running time are bounded by $O(\frac{k^*}{\delta^2\epsilon^2}\ln\frac{k^*}{\delta} + \frac{Hk^*}{\rho\delta^2})$, and $O(\frac{Hk^*}{\delta^2\epsilon^2}\ln\frac{k^*}{\delta} + \frac{H^2k^*}{\rho\delta^2})$, respectively.*

### B.1.2 Proof Technique

**Escape probability set size bounds.** To ease the presentation of the proof, we assume that $\widehat{\Delta}(F) = \Delta(F)$. This case is interesting by its own, since if the policy and transition function are known, we can compute $\Delta(F)$ efficiently using dynamic programming. As a result, event $\mathcal{E}$ always holds. In addition, it is clear that the termination of the algorithm implies that $\widehat{\Delta}(F) = \Delta(F) \leq 2\rho$, thus $F$ is $(2\rho + 2\epsilon)$−safe. The main challenge is bounding the size of $F$.

A few notations before we start. Let $F^*$ denote a minimal $\rho$−safe set (of size $k^*$). Consider iteration $i$ inside the while–loop. The random variable $G_i$ is the number of states in $F^*$ that are added to $F$ in iteration $i$ and $B_i$ is the number of states added to $F$ in iteration $i$ that are not in $F^*$ ($G$ stands for *good* and $B$ for *bad*). Notice that both $G_i$ and $B_i$ depend on the current set $F$. Notice that the size of the output set is exactly $\sum_i B_i + G_i$ and that $\sum_i G_i \leq k^*$.

The main idea of the proof technique is to show that by adding trajectories according to the new distribution $Q_F$, we ensure that, in expectation, there are at least as much good states that are added to $F$ as bad states. Suppose the trajectory $\tau$ was chosen to be added to $F^*$ by the algorithm. If $\tau \subseteq F^*$, then $G_i$ is equal to $new_F(\tau)$ and $B_i = 0$. If $\tau \nsubseteq F^*$, then $B_i \leq new_F(\tau)$. Summarizing these observations, we have the following bounds

$$G_i \geq new_F(\tau) \cdot \mathbb{I}[\tau \subseteq F^*] \text{ and } B_i \leq new_F(\tau) \cdot \mathbb{I}[\tau \nsubseteq F^*],$$

where $\mathbb{I}[\cdot]$ is the indicator function.

Moreover, a direct consequence of the probability in which $\tau$ is added to $F$ is that for any set of trajectories $T$ it holds that

$$\mathbb{E}_{\tau \sim Q_F}[new_F(\tau) \cdot \mathbb{I}[\tau \in T]] = \sum_{\tau \in T} Q_F(\tau) new_F(\tau)$$

$$= \frac{1}{Z} \sum_{\tau \in T, new_F(\tau) \neq 0} \left( \frac{\Pr[\tau]}{new_F(\tau)} \right) new_F(\tau) = \frac{1}{Z} \Pr_\tau[\tau \in T \wedge new_F(\tau) \neq 0], \tag{1}$$

where $Z$ is the normalization factor of $Q_F$.

To bound the size of $F$, we want to show that the algorithm does not add too many states outside of $F^*$. We therefore bound $\mathbb{E}[B_i]/\mathbb{E}[G_i]$, where the expectations are over the trajectory $\tau$ that is added to $F$ according to $Q_F$. Applying Equation (1) twice, once with $T = \{\tau \mid \tau \subseteq F^*\}$ and once with $T = \{\tau \mid \tau \nsubseteq F^*\}$, we bound the ratio between $B_i$ and $G_i$ by

$$\frac{\mathbb{E}[B_i]}{\mathbb{E}[G_i]} \leq \frac{\Pr_\tau[\tau \nsubseteq F^* \wedge new_F(\tau) \neq 0]}{\Pr_\tau[\tau \subseteq F^* \wedge new_F(\tau) \neq 0]}. \tag{2}$$

We know that $\Pr_\tau[\tau \nsubseteq F^*]$ is always smaller than $\rho$, so the numerator is $\leq \rho$. A lower bound for the denominator is

$$\Pr_\tau[new_F(\tau) \neq 0] - \Pr_\tau[\tau \nsubseteq F^*]. \tag{3}$$

Whenever the algorithm is inside the main loop, the safety is at least $\Pr_\tau[new_F(\tau) \neq 0] = \Delta(F) > 2\rho$. Thus (3) is lower bounded by $\rho$, and overall (2) is less or equal to 1, which implies that

$$\mathbb{E}[B_i] \leq \mathbb{E}[G_i]. \tag{4}$$

This completes the proof because we know that the algorithm does not add too many states outside of $F^*$. More precisely,

$$\mathbb{E}[|F|] = \mathbb{E}\left[ \sum_i B_i + G_i \right] \leq \mathbb{E}\left[ 2 \sum_i G_i \right] \leq 2k^*.$$

**Sample complexity.** To discuss the sample complexity, we drop the assumption that the MC is known to a learner, and uses *EstSafety* Subroutine to approximate $\Delta(F)$. The number of calls to *EstSafety* is bounded by the size of the output set, $F$. Hence, this part of the sample complexity is bounded by $|F| \cdot N_{|F|}$ and we show that is $O(\frac{k^*}{\epsilon^2} \log k^*)$. Another source of sampling is trajectories sampled for purposes of potentially adding them to $F$. Observe that at any iteration the set $F$ has escape probability of at least $2\rho$, and each trajectory that escapes $F$ is accepted with probability at

least $1/H$. This implies a lower bound for the probability that a random trajectory is accepted is $2\rho/H$. This gives an upper bound of $\frac{2|F|\rho}{H}$ for the expected sample complexity.

**Amplification.** Theorem B.1 shows that if $\mathcal{E}$ holds, then the set size, $|F|$, is bounded *in expectation* by $2k^*$. As $\Pr[\mathcal{E}] \geq 1 - \lambda$ implies, from Markov's inequality, that the size $(2 + \delta)k^*$ with small probability of about $\delta + \lambda = O(\delta)$. If we want to make sure that the actual size is at most $(2 + \delta)k^*$ with high probability, we can repeat the process about $\Theta\left(\frac{1}{\delta}\right)$ times and take the smallest size set.

For full proofs we refer to Appendix B.

## B.2   In-depth Analysis

For convince, we state here Hoeffding's inequality.

**Lemma B.3.** *[Hoeffding's Inequality] Let $y_1, \ldots, y_N$ be independent random variables such that $y_i \in [a, b]$ for every $y_i$ with probability 1. Then, for any $\epsilon > 0$,*

$$\Pr\left[\left|\frac{1}{N}\sum_{i=1}^{N} y_i - \mathbb{E}[y_i]\right| \geq \epsilon\right] \leq 2e^{-2N\epsilon^2/(b-a)^2}.$$

### B.2.1   Proof of Theorem B.1

In this section we provide a complete proof for Theorem B.1. Throughout the section, we define a few terms and notions. We will start with proving guarantees regarding a single iteration of the while–loop.

Recall that $F^*$ denotes a minimal $\rho-$safe set (of size $k^*$). If there are multiple optimal solutions, choose one arbitrarily. For the convince of analysis, we denote the values of the algorithm variables at the end of each iteration $i$ of the while–loop by $\tau_i, F_i, accept_i$. Let $j(i)$ denote the value of variable $j$ during the $i-$th call to *EstSafety* Subroutine. In addition, let $N_i$ denote the number of trajectories sampled for the $j-$th time of calling Subroutine *EstSafety*, i.e., $N_i = \frac{1}{2\epsilon^2} \ln \frac{2}{\lambda_{j(i)}}$ for $j(i) \leq i$.

For ease of presentation, we recall some of the definitions from the proof technique description. We say that a trajectory $\tau$ is *good* if all the states in $\tau$ are in $F^*$ and *bad* if it escapes it. I.e., a trajectory is good if $\tau \subseteq F^*$ and bad if $\tau \nsubseteq F^*$. Additionally, we say that a state $s \in \mathcal{S}$ is *good* if it is in $F^*$ and *bad* otherwise. Namely, a state $s$ is good if $s \in F^*$ and bad if $s \notin F^*$. Let $G_i(F_{i-1})$ and $B_i(F_{i-1})$ be the number of good and bad states added to $F_{i-1}$ in iteration $i$, respectively (notice that $G_i(F_{i-1})$ and $B_i(F_{i-1})$ are random variables that depends on $F_{i-1}$). For short, whenever it is clear from the context, we write $G_i$ and $B_i$ respectively.

The following lemma bounds the error in approximating the escape probability.

**Lemma B.4.** *Let $F_{i-1} \subseteq \mathcal{S}$ be a subset of of states and $\epsilon, \lambda_j > 0$ be some parameters. Let $S_i$ be a sample of $N_i \geq \frac{1}{2\epsilon^2} \ln \frac{2}{\lambda_{j(i)}}$ i.i.d. random trajectories. Then,*

$$\Pr_{S_i}\left[\left|\widehat{\Delta}(F_{i-1}) - \Delta(F_{i-1})\right| \geq \epsilon\right] \leq \lambda_j.$$

*Also, as $\lambda_j = \frac{3\lambda}{2(\pi j)^2}$,*

$$\Pr\left[\exists i \ \left|\widehat{\Delta}(F_{i-1}) - \Delta(F_{i-1})\right| \geq \epsilon\right] \leq \lambda/4,$$

*Where the last probability is over all the samples $S_i$ made by EstSafety Subroutine.*

*Proof.* The first part follows directly from Hoeffding's inequality by taking $y_i = \mathbb{I}[\tau \nsubseteq F]$.

Assigning $\lambda_j = \frac{3\lambda}{2(\pi j)^2}$ and applying union bound, we get

$$\Pr\left[\exists i \ \left|\widehat{\Delta}(F_{i-1}) - \Delta(F_{i-1})\right| \geq \epsilon\right] \leq \sum_i \Pr_{S_i}\left[\left|\widehat{\Delta}(F_{i-1}) - \Delta(F_{i-1})\right| \geq \epsilon\right]$$

$$\leq_{(*)} \sum_{j(i)} \lambda_{j(i)} \leq \sum_{j=1}^{\infty} \lambda_j = \sum_{j=1}^{\infty} \frac{3\lambda}{2(\pi j)^2} = \frac{\lambda}{4}.$$

The inequality marked by $(*)$ follows from the fact that $\Delta(F)$ is estimated once for every time $j$ increases. $\square$

We define the event that *EstSafety* always provides good estimations by
$$\mathcal{E} = \{\forall i \ \left|\widehat{\Delta}(F_{i-1}) - \Delta(F_{i-1})\right| \leq \epsilon\}.$$
By the above we have that $\Pr[\mathcal{E}] \geq 1 - \lambda/4$.

In the following lemma we assume that if the current escape probability is at least $2\rho$, then the fraction of bad trajectories that escape $F_{i-1}$ is bounded from above by the fraction of good trajectories that escape $F_{i-1}$.

**Lemma B.5.** *Let $\rho > 0$ and assume that $\Delta(F_{i-1}) \geq 2\rho$. Then,*
$$\Pr_\tau[new_{F_{i-1}}(\tau) \neq 0 \wedge \tau \not\subseteq F^*] \leq \Pr_\tau[new_{F_{i-1}}(\tau) \neq 0 \wedge \tau \subseteq F^*],$$
*where the probabilities are over random trajectories.*

*Proof.* To prove the lemma, we will bound the probability $\Pr_\tau[new_{F_{i-1}}(\tau) \neq 0 \wedge \tau \not\subseteq F^*]$ from above and the probability $\Pr_\tau[new_{F_{i-1}}(\tau) \neq 0 \wedge \tau \subseteq F^*]$ from below. Since $\Delta(F^*) \leq \rho$,
$$\Pr_\tau[new_{F_{i-1}}(\tau) \neq 0 \wedge \tau \not\subseteq F^*] \leq \Pr_\tau[\tau \not\subseteq F^*] \leq \rho. \tag{5}$$

The assumption $\Delta(F_{i-1}) \geq 2\rho$ implies that
$$2\rho \leq \Delta(F_{i-1}) = \Pr_\tau[new_{F_{i-1}}(\tau) \neq 0] = \Pr_\tau[new_{F_{i-1}}(\tau) \neq 0 \wedge \tau \subseteq F^*] + \Pr_\tau[new_{F_{i-1}}(\tau) \neq 0 \wedge \tau \not\subseteq F^*]$$
$$\leq \Pr_\tau[new_{F_{i-1}}(\tau) \neq 0 \wedge \tau \subseteq F^*] + \Pr_\tau[\tau \not\subseteq F^*] \leq \Pr_\tau[new_{F_{i-1}}(\tau) \neq 0 \wedge \tau \subseteq F^*] + \rho,$$
hence
$$\rho \leq \Pr_\tau[new_{F_{i-1}}(\tau) \neq 0 \wedge \tau \subseteq F^*]. \tag{6}$$
Putting (5) and (6) together yields the statement. $\qquad\square$

Now, as long as the algorithm is inside the while–loop (i.e., the escape probability holds $\widehat{\Delta}(F) > 2\rho + \epsilon$), it follows that $\Delta(F) \geq 2\rho$ with high probability from Lemma B.4. Combining it with Lemma B.5 would yield that with high probability over a random trajectory, if the trajectory escapes $F$ then in expectation it is at least as likely to be good as it is to be bad.

We move on to show the main ingredient of the proof, namely that for any iteration, with high probability, the expected number of good states added to the current set $F$ is larger or equal to the expected number of bad states.

For every iteration $i$ in which we sample $\tau_i$ both $G_i$ and $B_i$ depends on the following:

1. The realizations of the sampled trajectory, $\tau_i$, and in particular on $new_{F_{i-1}}(\tau_i)$.
2. The probability of adding it to $F$, i.e., $1/new_{F_{i-1}}(\tau_i)$.

Next, we prove Equation (4).

**Lemma B.6.** *Assume event $\mathcal{E}$ holds. Thus, for all iterations $i$ inside the while–loop we have*
$$\mathbb{E}[B_i|F_{i-1}] \leq \mathbb{E}[G_i|F_{i-1}],$$
*where the expectation is over the trajectory $\tau$ that is sampled from the MC dynamics and added to $F_{i-1}$ according to $Q_{F_{i-1}}$.*

*Proof.* Since event $\mathcal{E}$ holds, we have that $\Delta(F_{i-1}) \geq 2\rho$ as long as we do not terminate in iteration $i$.

We can use it to bound $\mathbb{E}_\tau[B_i|F_{i-1}]$ by
$$
\begin{aligned}
\mathbb{E}_\tau[B_i|F_{i-1}] \quad &\leq \quad \sum_{h=1}^{H} \frac{\Pr_\tau[new_{F_{i-1}}(\tau) = h \wedge \tau \not\subseteq F^*]}{h} \cdot h \\
&= \quad \Pr_\tau[new_{F_{i-1}}(\tau) \neq 0 \wedge \tau \not\subseteq F^*] \underbrace{\leq}_{Lemma\ B.5} \Pr_\tau[new_{F_{i-1}}(\tau) \neq 0 \wedge \tau \subseteq F^*] \\
&= \quad \sum_{h=1}^{H} \frac{\Pr_\tau[new_{F_{i-1}}(\tau) = h \wedge \tau \subseteq F^*]}{h} \cdot h \leq \mathbb{E}_\tau[G_i|F_{i-1}].
\end{aligned}
$$
$\qquad\square$

**Theorem B.1.** *Given $\rho, \epsilon, \lambda \in (0,1)$, FINDING SAFEZONE Algorithm returns a subset $F \subseteq \mathcal{S}$ such that:*

1. *The escape probability is bounded from above by $\Delta(F) \le 2\rho + 2\epsilon$, with probability $1 - \lambda$.*
2. *The expected size of $F$ given $\mathcal{E}$ is bounded by $\mathbb{E}[|F| \mid \mathcal{E}] \le 2k^*$.*
3. *The sample complexity of the algorithm is bounded by $O\left(\frac{k^*}{\lambda \epsilon^2} \ln \frac{k^*}{\lambda} + \frac{H k^*}{\rho \lambda}\right)$, and the running time is bounded by $O\left(\frac{H k^*}{\lambda \epsilon^2} \ln \frac{k^*}{\lambda} + \frac{H^2 k^*}{\rho \lambda}\right)$, with probability $1 - \lambda$.*

*Proof.* Assume that the event $\mathcal{E}$ holds, and recall that

$$\Pr[\mathcal{E}] \ge 1 - \lambda/4. \tag{7}$$

We start with the first clause. Since event $\mathcal{E}$ holds, Lemma B.4 in particular implies that $\Delta(F) \le 2\rho + 2\epsilon$, hence the first clause holds.

For second clause, we will bound $\mathbb{E}[|F| \mid \mathcal{E}]$ from above by $2k^*$. Since $\mathcal{E}$ holds, we have that $\Delta(F_{i-1}) \ge 2\rho$, for every $i$ inside the while–loop, thus Lemma B.6 yields

$$\mathbb{E}[B_i | F_{i-1}] \le \mathbb{E}[G_i | F_{i-1}].$$

This implies that

$$\mathbb{E}[|F| \mid \mathcal{E}] \le 2 \sum_i \mathbb{E}_{F_{i-1}}[\mathbb{E}[G_i | F_{i-1}]] | \mathcal{E}] \le 2k^*, \tag{8}$$

where the last inequality follows from the definition of $G_i$, as $\sum_i G_i \le |F^*| = k^*$.

We continue with the third clause of the theorem. Let $M$ denote the sample complexity of the algorithm, namely $M = M_F + M_E$ where $M_F$ is the expected total number of trajectories sampled within the FINDING SAFEZONE Algorithm (without the samples made by *EstSafety* Subroutine) and $M_E$ is total number of trajectories sampled using *EstSafety*. We will bound each term separately.

Since $\mathcal{E}$ holds, whenever we are inside the while–loop, $\Delta(F_i) \ge 2\rho$, which implies that it takes at most $1/2\rho$ trajectories in expectation to sample a trajectory that escapes $F_i$, and such trajectory is accepted with probability at least $1/H$. Thus, from Wald's identity, it follows that

$$\mathbb{E}\left[M_F | \mathcal{E}\right] = \frac{H}{2\rho} \cdot \mathbb{E}[|F| \,|\mathcal{E}] \le \frac{H k^*}{\rho}.$$

From Markov's inequality on the above inequality, with probability at least $1 - \frac{\lambda}{4}$,

$$\Pr\left[M_F \ge \frac{4H k^*}{\rho \lambda} \Big| \mathcal{E}\right] \le \frac{\lambda}{4}. \tag{9}$$

Moving on to bound $M_E$. Since $\mathcal{E}$ holds, it follows from Equation (8) and Markov's inequality that

$$\Pr\left[|F| \ge \frac{8k^*}{\lambda} \,\Big|\, \mathcal{E}\right] = \Pr\left[|F| \ge 2k^* \cdot \frac{4}{\lambda} \,\Big|\, \mathcal{E}\right] = \Pr\left[|F| \ge \mathbb{E}[|F| \mid \mathcal{E}] \cdot \frac{4}{\lambda} \,\Big|\, \mathcal{E}\right] \le \frac{\lambda}{4}. \tag{10}$$

If $|F| \le \frac{8k^*}{\lambda}$, the number of calls for Subroutine *EstSafety* is also bounded by $8\pi k^*/\lambda$ (we only call *EstSafety* after we added states to $F$). It also implies that $\frac{3\lambda^3}{2(8\pi k^*)^2} \le \lambda_j$ for every $j \ge 1$. Thus, if $|F| \le \frac{8k^*}{\lambda}$,

$$\begin{aligned}
M_E = \sum_{j=1}^{|F|} N_i &\le \sum_j^{\frac{8k^*}{\lambda}} \frac{1}{2\epsilon^2} \ln \frac{2}{\lambda_j} \le \sum_j^{\frac{8k^*}{\lambda}} \frac{1}{2\epsilon^2} \ln \frac{2}{\frac{3\lambda^3}{2(8\pi k^*)^2}} \le \sum_j^{\frac{8k^*}{\lambda}} \frac{1}{2\epsilon^2} \ln \frac{86(\pi k^*)^2}{\lambda^3} \\
&= \frac{8k^*}{2\lambda\epsilon^2} \ln \frac{86(\pi k^*)^2}{\lambda^3} = \frac{4k^*}{\lambda\epsilon^2} \ln \frac{86(\pi k^*)^2}{\lambda^3}
\end{aligned}$$

Combining the above with Equation (10), we get

$$\Pr\left[M_E > \frac{4k^*}{\lambda\epsilon^2} \ln \frac{86(\pi k^*)^2}{\lambda^3} \,\Big|\, \mathcal{E}\right] \le \frac{\lambda}{4} \tag{11}$$

As $M = M_F + M_E$, union bound over Equation (7), Equation (9) and Equation (11) implies that with probability $\geq 1 - 3\lambda/4 > 1 - \lambda$,

$$M = O\left(\frac{k^*}{\lambda\epsilon^2}\ln\frac{k^*}{\lambda} + \frac{Hk^*}{\rho\lambda}\right) \tag{12}$$

For each trajectory we sample we run in time $O(H)$, e.g., by using a lookup table for maintaining the current set $F$. Consequently, if the event in Equation (12) holds then the running time of the algorithm is bounded by

$$O\left(\frac{Hk^*}{\lambda\epsilon^2}\ln\frac{k^*}{\lambda} + \frac{H^2k^*}{\rho\lambda}\right).$$

Overall, all the clauses in the lemma hold with probability $\geq 1 - \lambda$.

$\square$

### B.2.2 Proof of Theorem B.2

**Theorem B.2.** *(main theorem) Given $\epsilon, \rho, \delta > 0$, if we run* FINDING SAFEZONE *for $\Theta(\frac{1}{\delta})$ times and return the smallest output set, $F \subseteq \mathcal{S}$, then with probability $\geq 0.99$*

1. *The escape probability is bounded by $\Delta(F) \leq 2\rho + 2\epsilon$.*
2. *The size of $F$ is bounded from above by $|F| \leq (2 + \delta)k^*$.*
3. *The total sample complexity and running time are bounded by $O(\frac{k^*}{\delta^2\epsilon^2}\ln\frac{k^*}{\delta} + \frac{Hk^*}{\rho\delta^2})$, and $O(\frac{Hk^*}{\delta^2\epsilon^2}\ln\frac{k^*}{\delta} + \frac{H^2k^*}{\rho\delta^2})$, respectively.*

*Proof.* Assume we run FINDING SAFEZONE Algorithm for $m = \frac{2\ln 300}{\delta}$ times and denote each algorithm output by $F^i$. Return the smallest set $F = \text{argmin}_{F^i}|F^i|$.

It follows from Theorem B.1 that for every $\lambda \in (0, 1)$, each $F^i$ is of expected size $\mathbb{E}[|F^i|] \leq 2k^*$, and is $(2\rho + 2\epsilon)$−safe with probability $\geq 1 - \lambda$. Choosing $\lambda = \frac{0.01}{3m}$ implies

$$\Pr[\Delta(F) > 2\rho + 2\epsilon] \leq \frac{0.01}{3}. \tag{13}$$

In addition, from Markov's inequality it follows that for every $\delta > 0$,

$$\Pr\left[|F^i| > (2 + \delta)k^*\right] \leq \Pr\left[|F^i| > (2 + \delta)k^*|\mathcal{E}\right] + \Pr[\mathcal{E}]$$
$$\leq \frac{2k^*}{(2 + \delta)k^*} + \lambda$$
$$= 1 - \frac{\delta/2}{1 + \delta/2} + \lambda$$
$$= 1 - \frac{\delta/2 - \lambda - \lambda\delta/2}{1 + \delta/2}$$

From the independence of the algorithm runs, for $m = \frac{2\ln 300}{\delta}$,

$$\Pr[|F| > (2 + \delta)k^*] \leq \Pr[\forall i : (|F^i| > (2 + \delta)k^*)]$$
$$\leq \prod_{i\in[m]}\Pr[|F^i| > (2 + \delta)k^*]$$
$$\leq \left(1 - \frac{\delta/2 - \lambda - \lambda\delta/2}{1 + \delta/2}\right)^m$$
$$\leq e^{-m\left(\frac{\delta/2 - \lambda - \lambda\delta/2}{1 + \delta/2}\right)} \leq \frac{0.01}{3}.$$

Hence

$$\Pr[|F| > (2 + \delta)k^*] \leq \frac{0.01}{3}. \tag{14}$$

As for the sample complexity, let $M_i$ denote the (random) sample complexity of the $i-$th run, and let us denote

$$\bar{M} = \frac{4k^*}{\lambda \epsilon^2} \ln \frac{86(\pi k^*)^2}{\lambda^3} + \frac{4Hk^*}{\rho \lambda}.$$

From Theorem B.1, $M_i > \bar{M}$ with probability $< \lambda$.

By taking union bound on the sample complexity bound per one run, we get

$$\Pr\left[\exists i : M_i > \bar{M}\right] \leq \sum_{i \in [m]} \Pr\left[M_i > \bar{M}\right] \leq m \cdot \lambda = \frac{0.01}{3}.$$

Where the last inequality follows from Theorem B.1, and $\lambda = \frac{0.01}{3m}$.

Assigning $m = \frac{2 \ln 300}{\delta}$ and $\lambda = \frac{0.01}{3m} = \frac{0.01\delta}{6 \ln 300}$, we get that with probability $\geq 1 - \frac{0.01}{3}$,

$$\sum_{i=1}^{m} M_i = O\left(\frac{mk^*}{\lambda \epsilon^2} \ln \frac{k^*}{\lambda} + \frac{mHk^*}{\rho \lambda}\right) = O\left(\frac{k^*}{\delta^2 \epsilon^2} \ln \frac{k^*}{\delta} + \frac{Hk^*}{\rho \delta^2}\right) \qquad (15)$$

Since the algorithm runs in time $O(H)$ for every trajectory sampled, if the sample complexity is bounded by the above term, then the total running time is bounded by $O\left(\frac{Hk^*}{\delta^2 \epsilon^2} \ln \frac{Hk^*}{\delta} + \frac{Hk^*}{\rho \delta^2}\right)$.

Finally, from union bound over Equation (13), Equation (14) and Equation (15) all the theorem properties hold with probability $\geq 0.99$. $\qquad \square$

