# OpenReview forum: "Finding Safe Zones of Markov Decision Processes Policies"
_NeurIPS.cc/2022/Workshop/TSRML — TSRML2022_

### Official Review · Reviewer_cKTA · 2022-10-21
**Developing a new notion of Safety for an MDP Policy parametrized by size and escape probability**

**Overall Rating:** 6

**Summary:**

The paper develops a new notion of safe zones of an MDP policy by evaluating the trajectory (random) generated by the policy with the number of states and escape probability. Any trajectory not within the safe zone of the policy is seemingly unsafe and needs evaluation. The ideal scenario is to obtain safe zones characterized by a lesser number of states and low escape probability which is basically the tighter conditions and naturally obtaining the optimal scenario is computationally hard. Hence, the authors obtain an approximate solution that gives a factor of almost 2 approximations with polynomial-size sample complexity.

**Strengths:**

The paper proposes an innovative definition of the safety of MDP policies which is a novel problem and can have possible downstream applications as demonstrated by the authors. Some of the use cases as motivated by the authors are quite interesting including finding the safe zone of a behavior policy and explaining the decision from a safety perspective. Identifying safety states can definitely tell if a trajectory violating the condition is unsafe and can provide counterfactual explanations to users as well. The proposed algorithm shows provable benefits over the past methods especially considering the size of safety states. The results are interesting and the paper is written in a clear manner.

**Weaknesses:**

Although the proposed approach shows improvement it can have computational challenges. On the other hand, the threshold and visitation-based approach is simplistic with provable benefits, why can't we add an additional constraint to the objective constraining the number of safety states, and trying to derive results with that is not very clear?
The notion of 'safety' is not extremely clear from how the definition of safety states have been provided. Although it is an interesting/novel way of defining it, still the safety aspect is not clear. Also, certain motivations seem to be force-fit, and not exactly clear how this new notion of safety applies there for ex: Approximate MDP equivalence. The paper lacks empirical evidence which would have been extremely helpful in highlighting the use-cases

**Overall Recommendation:**

An interesting way to define the safe zones of an MDP policy is with safety states and escape probabilities. The paper provides an interesting aspect of the problem and attempts to identify potential use cases where this alternate definition can be helpful and also derive an approximate solution to the problem with certain guarantees. However, it lacks a more clear definition explanation and empirical evidence to justify the potential of this notion of safety.

**Review Confidence:**

4: The reviewer is confident but not absolutely certain that the evaluation is correct

---

### Official Review · Reviewer_rymj · 2022-10-21
**Interesting problem with algorithms and their analyses to find a set of states deemed to be safe in an MDP although lacking in numerical examples**

**Overall Rating:** 7

**Summary:**

This paper introduces the problem of finding  SAFEZONE-a set of states in an MDP that are representative of states that occur in most of the trajectories. The SAFEZONE set ideally must have low escape probability $\rho$ (the probability that a state in a random trajectory is not included in the SAFEZONE) and a small cardinality $k$. The paper analyzes naïve algorithms and proposes an algorithm to combat the deficiency in naïve algorithms which is having a large set size for SAFEZONE.

**Strengths:**

The authors solve an interesting problem of trying to find the subset of ‘safe states’, i.e. states that occur in most trajectories and the probability that a random trajectory contains states other than those in the SAFEZONE is low.

To estimate SAFEZONE, the learner has access to a trajectory generator, total number of states in the Markov chain but no knowledge of the state transition function. These are general conditions under which the SAFEZONE problem is solved.

Apart from analyzing baseline algorithms to find SAFEZONE, they also propose an algorithm which identifies the SAFEZONE using sampled random trajectories in a way that the set-size k of SAFEZONE only grows gradually by accepting trajectories with a probability inversely proportional to the number of new states they add to SAFEZONE.


**Weaknesses:**


- Though a theoretical paper, it would benefit from a small synthetic dataset-based experiment just to give the reader an idea about the size of safe-zones for a simple policy.
- The optimal value of $k^*$ for a certain policy and an MDP can be very large depending on the problem, could the authors provide practical examples where $k^* \ll |\mathcal{S}|$?
- Where is the main theorem for which Theorem 2.2 is the ‘informal statement’?
- The paper is not very well organized especially Sections 3 and 4. All of them are algorithms to find the safe zone but from the section headings this is not clear. Also the text is interspersed with information in the appendix which makes the paper hard to read.
- Lines 168-169 on Page 4-please elaborate on the statement ‘while it is clear why..’
-References to equations and propositions are missing on Page 6.



**Overall Recommendation:**

It is a novel idea with complete analysis although lacking in some numerical examples to provide intuition to readers.

**Review Confidence:**

3: The reviewer is fairly confident that the evaluation is correct

---

### Decision · Program_Chairs · 2022-10-23

Accept